# Trends in Phase II Trials for Cancer Therapies

**DOI:** 10.3390/cancers13020178

**Published:** 2021-01-07

**Authors:** Faruque Azam, Alexei Vazquez

**Affiliations:** 1Wolfson Wohl Cancer Research Centre, Institute of Cancer Sciences, University of Glasgow, Garscube Estate, Switchback Road, Bearsden, Glasgow G61 1QH, UK; 2416474A@student.gla.ac.uk; 2Cancer Research UK Beatson Institute, Switchback Road, Bearsden, Glasgow G61 1BD, UK

**Keywords:** cancer, overall response rate, clinical trials, Phase II, drug combinations

## Abstract

**Simple Summary:**

Through time we have optimized drug combinations to treat cancer. Today we count with a better arsenal of cancer drugs acting on specific genes, known as targeted cancer therapies. Targeted therapies were promising as single drugs, avoiding the inevitable side effects of drug combinations. Here we determine whether that promise have been fulfilled. We collected data from thousand clinical trials testing the response of cancer drugs, either one drug at the time, or as part of a combination of drugs. We find that targeted therapies are better for the treatment of cancer when used in combination with previous cancer drugs that do not target specific genes. We conclude that drug combinations should continue as the standard of care for cancer therapy.

**Abstract:**

*Background*: Drug combinations are the standard of care in cancer treatment. Identifying effective cancer drug combinations has become more challenging because of the increasing number of drugs. However, a substantial number of cancer drugs stumble at Phase III clinical trials despite exhibiting favourable efficacy in the earlier Phase. *Methods*: We analysed recent Phase II cancer trials comprising 2165 response rates to uncover trends in cancer therapies and used a null model of non-interacting agents to infer synergistic and antagonistic drug combinations. We compared our latest efficacy dataset with a previous dataset to assess the progress of cancer therapy. *Results*: Targeted therapies reach higher response rates when used in combination with cytotoxic drugs. We identify four synergistic and 10 antagonistic combinations based on the observed and expected response rates. We demonstrate that recent targeted agents have not significantly increased the response rates. *Conclusions*: We conclude that either we are not making progress or response rate measured by tumour shrinkage is not a reliable surrogate endpoint for the targeted agents.

## 1. Introduction

Cancer treatment benefits from early detection as 90% of all cancer deaths occur at an advanced/metastatic stage [1]. The high mortality in advanced/metastatic disease is because of the unsatisfactory efficacy of currently available treatments, including targeted therapies. However, little progress has been made to inhibit metastasis owing to the poor understanding of the underlying metastatic process, infrequent use of preclinical metastatic models for drug screening and the complex tumour microenvironment [2]. Cancer metastasis follows a series of multicellular events involving interactions of neoplastic cells with non-cancerous stromal and immune cells of the tumour microenvironment [3]. These immune cells driven by the microenvironment modulate immune responses following cancer immunotherapy [4] and partly regulate chemotherapy sensitivity, and combinatorial treatment blocking tumour-associated macrophages has shown to enhance chemotherapy efficacy and restrict metastatic spread in transgenic breast cancer mouse models [5,6]. The rational integration of new targeted agents with cytotoxic drugs targeting the tumour and its microenvironment together could reduce cancer deaths significantly.

The influx of novel anticancer drugs along with existing chemotherapies poses a major challenge to the selection of effective drug combinations. The number of FDA-approved targeted therapies has increased five-fold compared to cytotoxic drugs [7]. Moreover, 63 distinct anticancer drugs were released on the market by the FDA between 2006 and 2016 [8], which would generate at least 39,000 different 3-agent combinations with an exponential growth. Unfortunately, the trend of trials testing combinatorial cancer therapies has lately decreased significantly relative to all oncology trials [9].

One important aspect of monitoring the trends of new cancer therapies is to minimise the high attrition rate of cancer drugs in Phase III trials. A recent comparative study reports that the success rate of cancer drugs is only 3.4%, whereas the overall success rate excluding oncology drugs is 20.9% [10]. Moreover, a few cancer drugs that pass-through Phase III trials do not always confer clinical benefit in the wider population. For instance, only one-third (45/133) of the single-arm trials supported by FDA-approval and 13 out of 37 released cancer drugs were translated to “meaningful clinical benefit” (MCB) according to the American Society of Clinical Oncology (ASCO) scales [8,11]. In addition, a combined analysis from two independent studies [12,13] investigating 243 randomised controlled trials (RCTs) of predominant cancers revealed that 36% (87/243) of the RCTs reached the minimum threshold of the MCB scale of the European Society for Medical Oncology (ESMO).

Post-market studies also point towards the incoherent performance of new cancer drugs between approval time and afterwards. Davis et al. [14] analysed 48 EMA-approved anticancer drugs and they found that most of the drugs did not extend survival or improve quality of life for a minimum of 3.3 years after market approval, although 35% of the indicated cancers were associated with significant survival benefits at approval-time. Likewise, Grössmann et al. [15] argue that approval status of a cancer drug does not represent MCB as most of the EMA-approved drugs between 2011–2016 had not reached ESMO’s MCB scale. Altogether, these discrepant post-approval performances of new cancer drugs in larger populations provide evidence towards the necessity to monitor trends and combination patterns of new cancer drugs before reaching Phase III trials.

The varying degrees of performances of targeted cancer agents in Phase III trials have been rendering the trends more difficult to study. This is possibly due to the surrogate endpoints, overall response rate (ORR) and progression-free survival (PFS), used in earlier trials that are not sufficient to predict the overall survival (OS). In concordance with this, several analyses [16,17,18,19,20,21] highlight that improved ORR or longer PFS do not always correlate to survival benefit, and there are often little or unknown correlations between surrogate endpoints and OS. Undoubtedly, targeted cancer therapies have impacted the treatment outcome profoundly, although effective only in a small cancer subpopulation with specific biomarkers, while chemotherapy has made a modest difference across all the stages of disease in all population [22].

The question regarding the superiority of the targeted agents over chemotherapies is disputable. This dichotomy has resurfaced from the failure of the targeted agents to deliver a survival benefit even in biomarker-specific subset of population. For instance, Camidge emphasised that the majority number of Phase III studies of tyrosine kinase inhibitors (TKIs) testing EGFR-mutated non-small cell lung cancer (NSCLC) patients could not demonstrate OS superiority over chemotherapy regardless of the significant ORR and PFS improvement [22]. However, multiple Phase II and III studies [23,24,25,26,27,28,29] of HER2+ metastatic breast cancer proved that rational combinations of chemotherapies to targeted agents are more safe and effective. In retrospect, owing to all incongruent results of targeted agents, it is one of the clinical unmet needs to understand how novel cancer drugs are performing in Phase II trials and analyse them in large numbers to detect small differences and recognise the pattern of synergy/antagonism for prospective Phase III trials.

Combinatorial therapy in metastatic disease can deliver key advantages over monotherapy given the complex interactions of the tumour immune microenvironment [6]. It allows combination of multiple biologically distinct drugs to gain superior activity over monotherapy by enhancing pharmacodynamic activity through synergy, overcoming the resistance problem, reducing the required concentrations of each combined agent, and minimising the dose-dependent toxicity [30,31,32,33]. Furthermore, it is well known that combination chemotherapy results in better efficacy and response rate compared to monotherapy, although as explained above for targeted therapies, the role of combination therapy on overall survival remains ambiguous [34].

In the search for effective cancer drug combinations, a balanced approach is to analyse a large number of Phase II trial data to monitor trends of new cancer drugs and understand the response pattern and interactions, thus identify potential synergistic and antagonistic combinations. Moreover, Phase II trials have a reasonable number of study participants as opposed to a very little participants in Phase I trials. On the other hand, there are considerably a greater number of Phase II trials available to study than Phase III trials. Meta-analyses and pooling together a large number of clinical data have been analysed to assess the efficacy of novel cancer drugs against standard treatments [35,36,37,38]. Hence, interpretations from bulk clinical data could potentially shed light on the current hazy situation rendered by the abundant choices of cancer drugs.

In this study, we accumulated 2165 Phase II trials’ ORR data covering three decades and identified a trend of cancer drugs, inferred synergistic and antagonistic combinations, and also explored how the trends of cancer treatments have changed over time.

## 2. Experimental Methods

To investigate the trends in cancer combination therapy we have collected ORRs from Phase II clinical trials (Figure 1).

### 2.1. Endpoint Clinical Variable

The overall response rate (ORR) in a clinical trial is defined as the total percentage of patients achieving a complete and partial response after treatment. A complete response refers to the patients whose tumour disappeared after treatment and a partial response generally refers to the patients achieving a predefined reduction (usually ≥30%) in the target lesions or tumour volume or cell number.

### 2.2. ORR Data Source and Selection Criteria

The ORR data were collected from PubMed. On 15 April 2020, data were searched with the following: cancer Phase II clinical trial overall response rate. From the returned list of abstracts, 1002 ORR data were extracted, starting from Phase II clinical trials of the year 2020 and onwards as they appeared in order. The collected ORR data in a clinical trial consisted of the patients who were evaluable for tumour response after treatment, excluding the intention-to-treat population ORR data. In some cases where the ORR was not directly specified, the ORR was manually calculated by combining complete and partial response data from the efficacy result or supplementary data. Clinical trials that did not have the ORR as primary or secondary endpoint were disregarded. In our collected dataset, the ORRs of trials testing non-targeted agents (cytotoxic) in solid tumours was assessed by Response Evaluation Criteria in Solid Tumours (RECIST v1) [39] and the RECIST v1.1 [40] was used for targeted agents. For haematological malignancies, Lugano response criteria [41] and International Working Group’s revised response criteria [42,43] were used to assess the ORR for lymphomas and leukaemia. On 20 June 2020, a total of 1002 Phase II clinical trials with response data comprising of 44,429 subjects were compiled in a spreadsheet for subsequent analysis (Appendix A).

### 2.3. Agent Classification

Conventional chemotherapeutic and cytotoxic drugs were classified as non-targeted agents. In contrast, synthetic hormonal therapies targeting specific receptor or receptors, monoclonal antibodies, molecularly targeted cancer drugs such as small molecule kinase inhibitors, and modern immunotherapies including checkpoint blockers and CAR-T cell were classified as targeted agents (Table 1). Radiation and surgical interventions to regress tumour, and steroids such as prednisone and dexamethasone which were used to alleviate tumour associated pain and swelling had been neither deemed as non-targeted nor targeted agents but counted as an agent in the analyses of combinations.

### 2.4. Statistical Analysis

When two groups were compared for a difference in mean ORR, all the performed statistical tests were two-tailed Student’s *t*-test at 5% significance level. Bonferonni correction was employed when simultaneous significance tests had been done within the same ORR groups in order to minimise the experiment-wise error rate.

### 2.5. Clinical Synergy and Antagonism

Clinical synergy and antagonism for combinations were calculated using a null model of non-interacting agents, which was postulated by Kang et al. [38]. This model accounts for synergy or antagonism of drugs based on observed ORR and expected ORR of a combination, while assuming no interactions between the agents. Evidence of synergy was found when the observed ORR of a drug combination was significantly greater (*p*
_synergy_ < 0.05) than the expected ORR. In contrast, evidence of antagonism was found if the observed ORR was significantly lesser (*p*
_antagonism_ < 0.05) than the expected ORR. The expected ORR for a combination consisting of drug A and drug B was calculated by the following equation:ORR_expected_ = 100% [1 − (1 − ORR_A_/100%) (1 − ORR_B_/100%)],(1)
where ORR_A_ and ORR_B_ correspond to the mean ORR from the trials testing drug A and drug B as single-agent, respectively. Consequently, the observed ORR for the combination (drug A + drug B) was all the ORRs from trials testing drug A and drug B together.

## 3. Results

### 3.1. Impact of Combination Size

The ORRs are reported in Figure 2a,b, binned according to the number of drug combinations in the clinical trials. The ORR started from 29% for clinical trials testing a single-agent and significantly increased to reach 54% for 3-agent combinations (Figure 2b). For trials testing 4- and 5-agent combinations the ORRs did not significantly exceed the ORR of 3-agent trials (Figure 2b). For trials testing 6- or 7-agent combinations the average ORR exhibited wide variations (Figure 2b). First, the ORR goes up by almost a 30% from trials testing 3–5 agents to 6-agent trials. Then the ORR drops down by a 45% from trials testing six agents to 7-agent trials. These wide variations are most likely due to the low number of reported trials testing six and seven agents (Figure 2a). In the following we restrict our attention to trials testing 1–5 combinations. Finally, when we restrict the analysis to trials testing at least one targeted agent, we observe the exact same trends with slightly better ORRs for 4- and 5-agent combinations (Figure 2c).

### 3.2. One versus Multiple Targeted Agents

The data shown above indicate that, on average, increasing the number of agents increases the ORR. It is worth asking if increasing the number of targeted agents will give an advantage compared to adding non-targeted agents. To address this question, we compared clinical trials with the same number of agents but stratified into having one (single) or more than one (multiple) targeted agents. Overall, we did not observe a clear improvement in the ORR of multiple-targeted agents when compared to corresponding single-targeted agent combinations (Figure 3). For example, the ORR of 2-agent single-targeted agents (one targeted plus one non-targeted agent) was significantly higher (46% vs. 35%) than two targeted agents combined. Conversely, in 4-agent combinations, the ORR of one targeted plus three non-targeted agents was significantly lower (54% vs. 72%) than two targeted plus two non-targeted agents. These data suggest that the combination of targeted agents has not been sufficiently optimized for non-targeted agents.

### 3.3. Trends across Time

To analyse the trends in cancer therapy, we compared the current results (2013–2020) with a dataset from a previous study [38] covering Phase II clinical trials between the year 1990–2011 (modern vs. previous). As expected, the modern dataset contains an increased proportion of targeted agents when compared to the older dataset (Figure 4a). Overall, except for the 2-agent combinations, we do not observe significant differences between the modern and previous trends of the ORR as a function of the number of agents (Figure 4b). There are some variations for 5-agent combinations but, these are probably due to a particular 5-agent combination with a low ORR appearing six times on that bin which skewed down the mean ORR (see discussion). Unexpectedly, the enrichment with targeted agents in modern Phase II trials is not translated into an average increase in the ORR.

### 3.4. Synergistic and Antagonistic Combinations

Synergy and antagonism of drug combinations can be estimated using a null model that assumes no interactions between agents [38]. A combination is deemed synergistic if the observed ORR (ORR_O_) from the clinical trials of that combination significantly exceeds the expectation from the null model of non-interacting agents (ORR_E_). Likewise, a combination is deemed antagonistic if the ORR_E_ significantly exceeds ORR_O_. The application of this methodology to evaluable Phase II trial data uncovered several synergistic (ORR_O_ > ORR_E_, *p*
_synergy_ < 0.05) and antagonistic (ORR_O_ < ORR_E_, *p*
_antagonism_ < 0.05) combinations (Figure 5, Table 2 and Table 3).

We identified four synergistic and 10 antagonistic combinations and plotted them in Figure 5. The diagonal line represents no discrepancy between observed and expected ORR. Consecutively, the distance of a combination from their corresponding position in the diagonal line i.e., straight upward for synergy and downward for antagonism, resembles the degree of difference between the observed and expected ORR.

Of the four synergistic pairs (Table 2), three pairs consist of chemotherapies, while the remaining pair contains two targeted agents (rituximab + ibrutinib). Notably, either carboplatin or nab-paclitaxel appear in all three of the synergistic chemotherapy combinations.

Of the 10 antagonistic drug combinations (Table 3), seven of them consist of at least one targeted agent and four of the drug pairs contain two targeted agents: one monoclonal antibody and one tyrosine kinase inhibitor. Notably, among the six antagonistic chemotherapies, gemcitabine is associated with four of the combinations. Moreover, among the five antagonistic drugs FOLFOXIRI (leucovorin, fluorouracil, oxaliplatin, irinotecan) and cetuximab, the interaction between irinotecan and cetuximab combination was identified as antagonistic independently (See discussion).

## 4. Discussion

We observed varying degrees of ORR trends of cancer drugs depending on the types and number of agents in combinations and also inferred four synergistic and 10 antagonistic combinations. Targeted agents clearly demonstrated superior efficacy over non-targeted cytotoxic agents in our dataset. However, one targeted agent with one non-targeted agent significantly produced better efficacy than two targeted agents combined. Unexpectedly, the comparison of the modern dataset with the previous efficacy dataset revealed no significant increase in the ORR trend of the targeted agents in recent trials.

In our analysis, the ORR trends of targeted agents (Figure 2c) and all cancer agents (Figure 2b) followed a similar increasing trend with no discernible differences. However, 4-agent and 5-agent combinations of targeted agents exhibited a slightly higher ORRs than all cancer agents. This indicates that targeted agents perform relatively better in four to five drugs’ combination than corresponding lower combination sizes. In light of this finding, replacing a targeted agent by a non-targeted agent is proven to be optimal in a combination of two targeted agents (Figure 3).

We suggest that recent targeted agents are not optimised properly in chemotherapy combinations. To demonstrate, the addition of panitumumab [44] and cetuximab [45] to bevacizumab-chemotherapy combinations in metastatic colorectal cancer (mCRC) RCTs reduced PFS and OS, and found to be suboptimal. Many promising targeted agents stumble in clinical trials despite a favourable preclinical profile. In line with this, a recent umbrella trial assessing precision medicine in NSCLC exposes that most of the investigational single-targeted agents have shown poor response rates (<10%) and few treatment cohorts have been discarded because of insufficient efficacy, whereas response rates were much higher for double-targeted agents [46]. Moreover, targeted agents’ performance is difficult to predict in a wider drug-biomarker specific subpopulation. For instance, two randomised Phase III trials suggest that afatinib failed to prolong patient life in the whole tested population of EGFR-mutant advanced lung cancer [47], however, afatinib significantly extended survival by 3 months to a specific EGFR-mutated subgroup compared to chemotherapy [48].

To make matters more complicated, inconsistent performances among different generations of EGFR-TKIs have been noticed when multiple trials’ results are being analysed. A randomised controlled Phase II trial assessing the performance of first-generation (gefitinib) and second-generation (afatinib) EGFR-TKIs revealed a significant improved PFS of afatinib in EGFR-mutant NSCLC [49]. However, a recent network meta-analysis of eight studies has identified that gefitinib is associated with longer OS than afatinib despite displaying a shorter PFS in EGFR-mutant NSCLC brain metastasis [21]. Likewise, Camidge argued that TKIs in NSCLC do not considerably extend patient survival while conferring a better PFS and ORR at the initial Phases [22]. However, this transient benefit simply reallocates the total available survival time compared to historical chemotherapy data. Although it is undeniable to overlook targeted agents’ profound impact on overall survival benefit but all of these studies indicate toward investigation for more specific and actionable biomarkers of targeted agents [2,22].

Surprisingly, we observed that the ORR trend in our modern dataset is relatively lower than the previous dataset, which reflects no treatment improvements over time. However, an alternative explanation of this incongruous trend could be the insufficiency of ORR as an endpoint to evaluate targeted agents. In our dataset, the response rate of the targeted agents in solid tumour trials was largely assessed by the RECIST 1.1 [40], while the previous version (RECIST v1.0) was used for trials of cytotoxic drugs. This is because the RECIST v1.0 [39] was originally developed to assess the efficacy of cytotoxic drugs.

The RECIST is based on tumour shrinkage and involves unidimensional radiographic measurement of target lesions. Multiple studies [50,51,52,53] have suggested that tumour size reduction may not always be symmetrical, especially for targeted agents because of their mechanisms which do not regress tumour by cytotoxicity, and complex tumour microenvironments. Furthermore, several retrospective studies [52,54,55,56,57] evidence toward bevacizumab’s superior pathological response than chemotherapy regardless of the similar RECIST response rates, and suggest that pathological response defined by the cell’s morphological change could be a better predictor of OS for preoperative chemotherapy in colorectal hepatic metastases [56,58,59]. However, more precise non-invasive methods for determining pathological response rate need to be developed.

We suggest that the response rate of targeted agents measured by the RECIST method might not be a reliable surrogate endpoint for overall survival. In line with this, two independent Phase III studies [60,61] have reported that cetuximab and bevacizumab do not improve RECIST-defined ORR significantly when combined with standard chemotherapy regimens in mCRC, however, the addition of bevacizumab significantly prolonged PFS but failed to extend OS and ORR, whereas cetuximab extended OS without changing the ORR and PFS. This implies that ORR is incapable of predicting the OS for bevacizumab and cetuximab, and no concordance between ORR and PFS. Meta-analysis combining three Phase III trials of metastatic breast cancer consisting of 2695 subjects revealed that bevacizumab significantly enhanced ORR and PFS when added to chemotherapy, although this increase did not reflect into significant OS benefit [36]. Therefore, all of these discrepant studies point toward the failure of the RECIST response rate as an indicator of patient benefit for targeted agents in mCRC [62] and breast cancer.

As mentioned in Section 3, the ORR of the 5-agent trials is likely to be outliers because largest ORR differences were originating from it. Besides, the ORRs data from 5-agent to 7-agent trials itself had been less reliable as the number of those trials in our dataset decreased dramatically for the higher number of combinations. However, the numbers of 5-agent and 6-agent trials are doubled and more than tripled, respectively, in our dataset (modern) compared to the older dataset (previous). After comparing the modern dataset with the previous (Figure 4b), we expected our 5-agent combinations’ ORR to be relatively higher. Therefore, a closer look into the lowest ORRs within 5-agent combination trials uncovered an unusual combination appearing six times. The suspected 5-agent combination consisted of celecoxib, thalidomide, fenofibrate, cyclophosphamide, and etoposide, and the mean ORR was only 6.75%, ranging from various CNS tumours to bone cancer trials. This specific combination skewed down the 5-agent trials’ ORR. On the other hand, we tried to identify which agents had contributed to the high ORR of the 6-agent combinations. Two specific combinations containing three distinct targeted agents, venetoclax plus obinutuzumab and venetoclax plus rituximab, partly contributed to the heightened ORR of 6-agent combinations because of their frequent appearance in those trials.

We found the combination of cetuximab and FOLFOXIRI (leucovorin, fluorouracil, oxaliplatin, irinotecan) chemotherapy regimen antagonistic in mCRC. Moreover, we identified that the combination of cetuximab and irinotecan itself was antagonistic in mCRC (Table 3), which further substantiates the antagonism between cetuximab and FOLFOXIRI. However, using the same methodology, Kang et al. [38] found that oxaliplatin and irinotecan combination was synergistic in colorectal cancer, implying that at least one synergistic and one antagonistic two-drug interactions exist between the five drugs. This finding is relevant in light of the results from clinical trials where cetuximab, bevacizumab, and panitumumab were somewhat not recommended and subject to careful addition to oxaliplatin or irinotecan based chemotherapy regimens in mCRC patients [44,45,60]. Looking in our synergistic drug pairs (Table 2), we identified doxorubicin and carboplatin combination was synergistic in ovarian cancer. In line with this, Kang et al. [38] inferred a similar but not identical combination, doxorubicin and oxaliplatin, to be synergistic in ovarian cancer.

There are caveats associated with the inferred synergy and antagonism. Firstly, the identification of a particular synergistic/antagonistic combination was restricted by the availability of the trials testing that combination and their respective single-agent trials in our dataset. Secondly, the null model would not account for drugs that are not mutually exclusive such as drugs with similar mechanisms of actions interacting with each other [63]. Thirdly, varying degrees of synergy/antagonism of the inferred combination would be expected in vitro at different dose-ratio. This is because the shape of the dose-effect curve of the inferred combination depends on the specific dose-ratio used in those trials in our dataset. Fourthly, a significant greater combined effect does not necessarily indicate synergy, which can result from additive effects or even a minor antagonism [64]. Therefore, synergy needs to be verified and quantified in vitro by Chou-Talalay’s method [63] to understand mechanisms and extent of conferred synergy of each agents in combination.

Our analysis does not apply to a specific cancer type for a given combination, rather it was focused on a macro-level to explore overall trends of new cancer drug combinations. However, results relating to a specific molecularly targeted agent would likely applicable to specific cancer subtypes, i.e., trastuzumab for HER2+ breast and stomach cancer. Reflecting on the response rate endpoint, it is not clear as to whether an increased ORR conferred by the targeted agents translates into a survival benefit, or the ORR itself measured by tumour shrinkage is not representing the true performance of the targeted agents. However, it is reasonable to conclude that the ORR of targeted agents is not a reliable surrogate endpoint for OS. Nonetheless, our analysis could be influenced by publication bias as trials with negative outcomes would more likely to remain unpublished. Altogether, our findings will provide insight on how new cancer drugs are performing in general and the need for optimising them in combinatorial therapies.

## 5. Conclusions

Our analysis demonstrates that targeted therapies should be used in combination with cytotoxic drugs to reach high response rates. We identified four synergistic and 10 antagonistic combinations based on the observed and expected response rates. The comparison of recent and older collections of clinical trials does not manifest signs of improvements in the overall response rate. Recent targeted agents have not significantly increased the response rates. We conclude either we are not making progress or response rate measured by tumour shrinkage is not a reliable surrogate endpoint for the targeted agents.

## Figures and Tables

**Figure 1 cancers-13-00178-f001:**
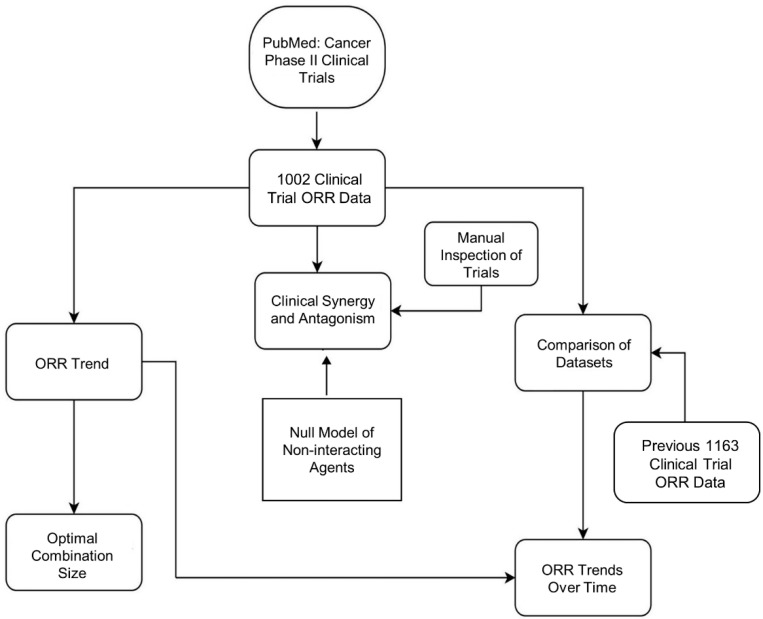
Study design and workflow. Previous 1163 ORR and null model from Kang et al. [38].

**Figure 2 cancers-13-00178-f002:**
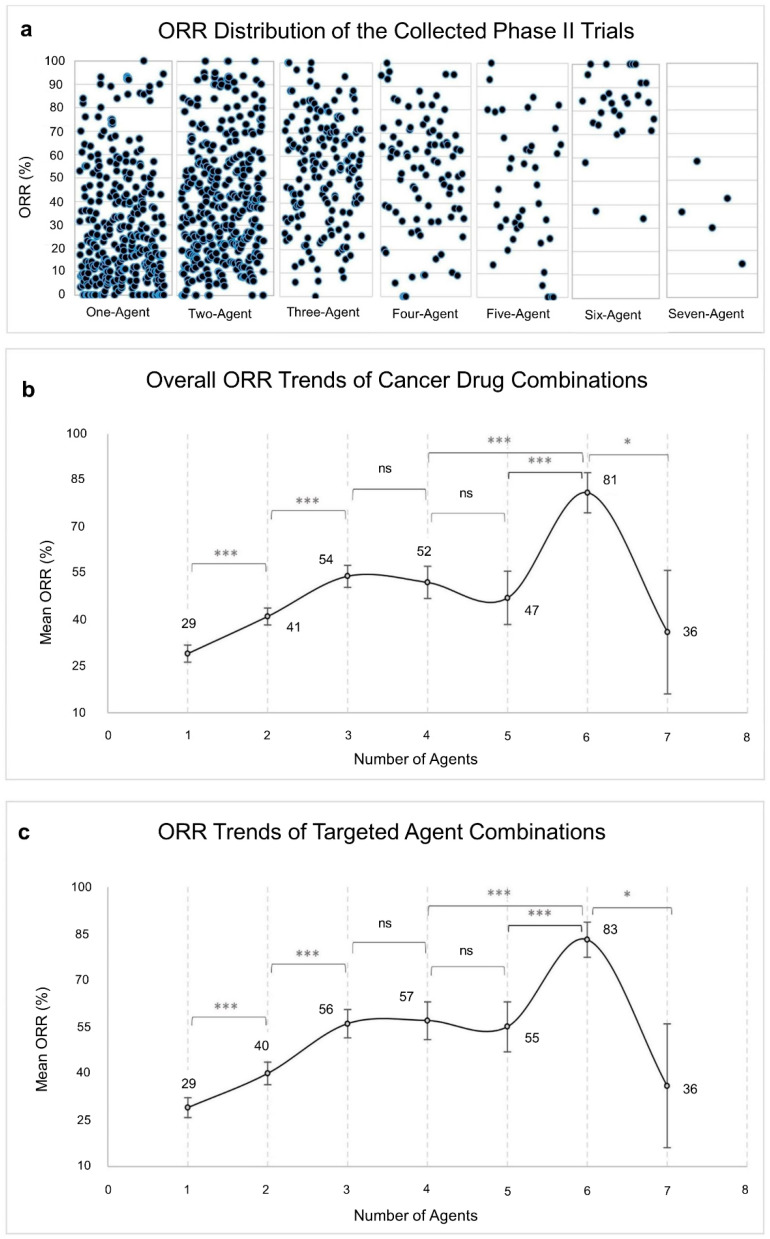
ORR increases with increasing number of agents in combination. (**a**), Distribution of the collected Phase II trials’ ORRs according to combination size. (**b**), ORR trends of all cancer drug combinations of the collected Phase II trials. (**c**), ORR trends of the targeted agents in combination with non-targeted agents, excluding trials with no targeted agents, *n* = 721. Points and error bars represent the mean ORR and 95% confidence interval, respectively. Data were analysed by two-tailed Student’s *t*-test with Bonferroni correction. * *p* < 0.007, *** *p* < 0.0001.

**Figure 3 cancers-13-00178-f003:**
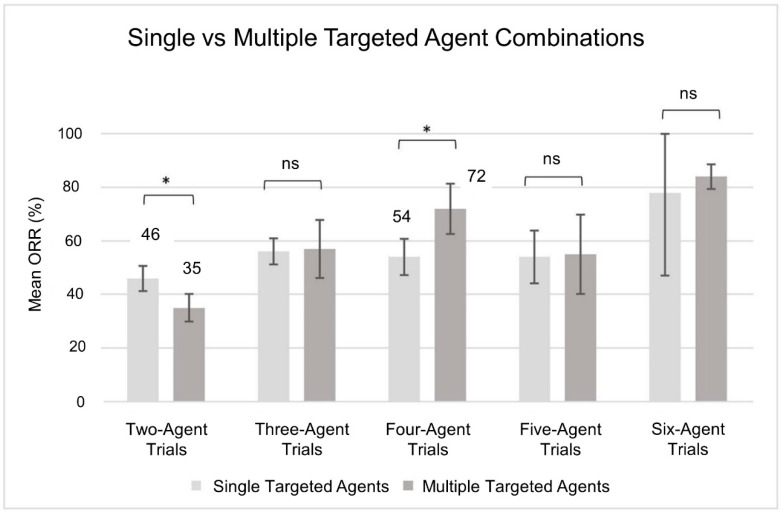
Increasing the number of targeted agents does not increase ORR. Single and multiple-targeted agent combinations contain one and more than one targeted agents, respectively, with or without non-targeted agents. Single-targeted agent trials, *n* = 290 and multiple-targeted agent trials, *n* = 167. The bars and error bars represent the mean ORRs and 95% confidence interval, respectively. The statistical significance was estimated by two-tailed Student’s *t*-test, * *p* < 0.05.

**Figure 4 cancers-13-00178-f004:**
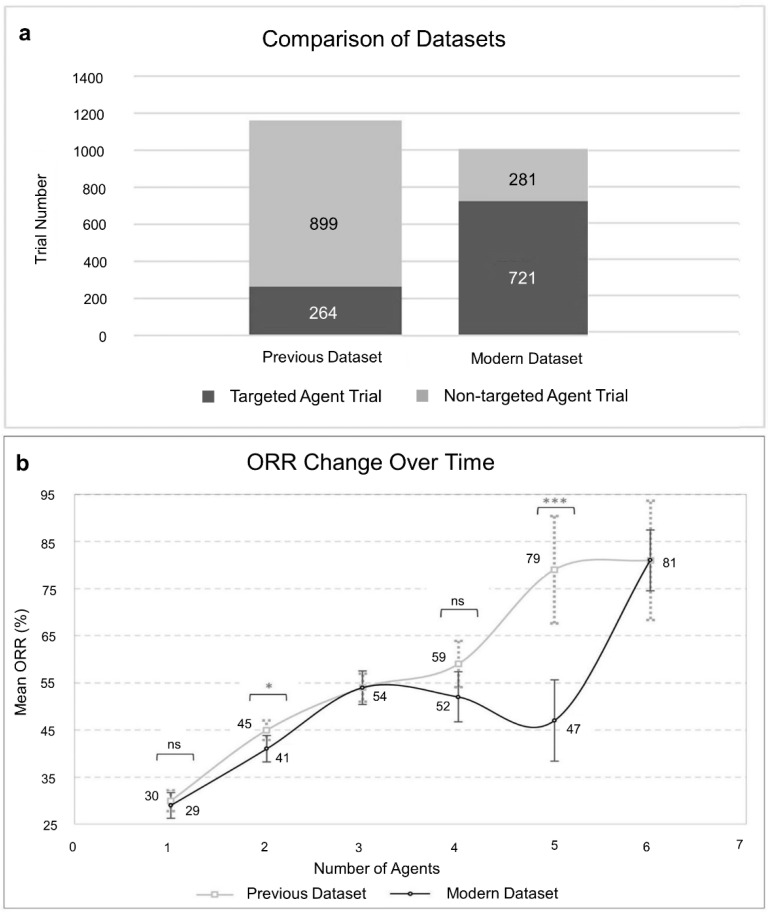
ORR trends over time. (**a**)**,** Proportion of targeted and non-targeted agent trials in the previous and modern ORR datasets. (**b**)**,** ORR as a function of the number of agents according to the combination tested in the previous and modern datasets. Previous dataset, *n* = 1163 and modern dataset, *n* = 1002. Points and error bars represent the mean ORR and 95% confidence interval, respectively. The statistical significance was estimated by two-tailed Student’s *t*-test, * *p* < 0.05, *** *p* < 0.001. Previous dataset from Kang et al. [38].

**Figure 5 cancers-13-00178-f005:**
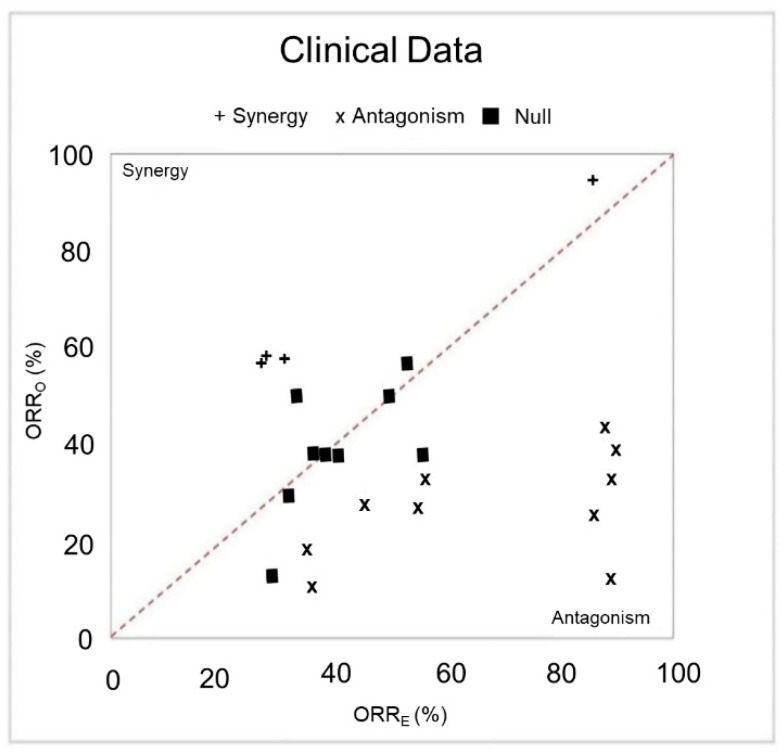
Clinical synergy and antagonism. The observed ORRs (ORR_O_) as a function of the expected ORR (ORR_E_) assuming no agent-agent interactions (null model). The diagonal line represents the perfect agreement with the null model. The left side and right side of the diagonal line correspond to the region of synergy and antagonism, respectively. (+) denotes combinations having evidence for synergy: ORR_O_ > ORR_E_, *p*
_synergy_ < 0.05; (×) combinations having evidence for antagonism: ORR_O_ < ORR_E_, *p*
_antagonism_ < 0.05; and black squares (■) no significant difference from the null model.

**Table 1 cancers-13-00178-t001:** Targeted and non-targeted agent classification.

Drug Class	Targeted Agents	Non-Targeted Agents
Cytotoxic drugs		Doxorubicin, Cisplatin, Nab-Paclitaxel
Synthetic hormonal agents	Abiraterone, Fulvestrant, Anastrozole	
Monoclonal antibodies	Bevacizumab, Trastuzumab, Rituximab	
Tyrosine kinase inhibitors	Sunitinib, Ibrutinib, Erdafitinib	
Proteasome inhibitors	Bortezomib, Carfilzomib, Ixazomib	
Modern immunotherapies	Pembrolizumab, Nivolumab, CAR-T cell	
Other	Interleukin-2, Everolimus, Temsirolimus	Pomalidomide, Lenalidomide

**Table 2 cancers-13-00178-t002:** List of inferred synergistic combinations.

Synergistic Combinations						
Agent 1	Agent 2	Expected ORR_E_ (%)	Observed ORR_O_ (%)	*p* _synergy_	Cancer Subtype	Null Model
Doxorubicin	Carboplatin	27	58	9.33 × 10^−3^	Ovarian cancer	Kang et al. [38]
Carboplatin	Nab-Paclitaxel	28	59	4.87 × 10^−3^	Lung (NSCLC *), Oropharyngeal, Breast cancer (TNBC **)	
Rituximab	Ibrutinib	86	94	1.71 × 10^−3^	Chronic lymphocytic leukaemia	
S-1	Nab-Paclitaxel	31	58	2.59 × 10^−2^	Gastric, Pancreatic cancer	

* Non-small cell lung cancer. ** Triple negative breast cancer.

**Table 3 cancers-13-00178-t003:** List of inferred antagonistic combinations.

Antagonistic Combinations						
Agent 1/Combination 1	Agent 2	Expected ORR_E_ (%)	Observed ORR_O_ (%)	*p* _antagonism_	Cancer Subtype	Null Model
Afatinib	Bevacizumab	35	18	1.88 × 10^−2^	Lung cancer (NSCLC *, EGFR Mutant)	Kang et al. [38]
Carboplatin	Gemcitabine	88	43	5.33 × 10^−3^	Ovarian, Breast (TNBC **), Lung cancer (Squamous NSCLC *)	
Ibrutinib	Durvalumab	86	26	1.30 × 10^−3^	Non-Hodgkin lymphoma	
Erlotinib	Bevacizumab	36	10	1.67 × 10^−4^	Hepatocellular carcinoma	
Erlotinib	Gemcitabine	89	13	4.96 × 10^−3^	Metastatic pancreatic cancer	
Nab-Paclitaxel	Gemcitabine	88	33	8.87 × 10^−6^	Pancreatic, Breast, Bile duct cancer	
Gemcitabine	Paclitaxel	89	39	3.87 × 10^−2^	Metastatic breast cancer	
Trastuzumab	Neratinib	54	27	3.27 × 10^−2^	Breast cancer (HER2+) ***	
Irinotecan	Cetuximab	45	28	1.94 × 10^−2^	Metastatic colorectal cancer (KRASwt, BRAFwt) ****	
FOLFOXIRI ^#^	Cetuximab	56	34	4.00 × 10^−3^	Metastatic colorectal cancer	

* Non-small cell lung cancer. ** Triple negative breast cancer. *** Human epidermal growth factor receptor 2. **** Wild type KRAS, wild type BRAF. ^#^ Leucovorin + Fluorouracil + Oxaliplatin + Irinotecan.

## Data Availability

All data is included within the submission.

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
