# Peer review of "Trends in Phase II Trials for Cancer Therapies"

_cancers, 2021, doi:10.3390/cancers13020178_

Round 1
Reviewer 1 Report
In this review, Azam and Vazquez propose an overview of Phase II clinical trial for cancer therapies of the past ten years with a focus on how the combination of targeted and non-targeted agents could be more effective than the combination of same-type drugs.
Abstract and discussion are clear and well written and promises well for the rest of the paper. Unfortunately, results section is only drafted and conclusion is confused and too long.
Following comments are meant to improve the paper and make it more comprehensible.
Major issues:
- lines 123-124: authors declare to have used/calculated ORR based on RECIST which is a method (as declared in these lines by authors) relative to solid tumors. This implies that haematological cancers are not taken into consideration in this article. Maybe, this point needs to be clearly stated. Moreover, in Table 1, are cited Chronic Lymphocytic Leukemia and Non-Hodgkin Lymphoma. In these cases, probably RECIST criteria are referred to lymph nodes, but a clarification could be useful
- line 190: authors constantly refer to a previous study (in this line and followings) but never provide a reference. It is mandatory to provide it
- line 195: “… due to the lack of clinical data...”, actually the number of the reported studies with 5 agents are not so limited but the real problem could be the large difference between the old study and the modern data since they are, respectively 79 and 47. Likewise, trials with 6 agents are 81 in both studies so they are not limited nor different, nevertheless, no differences in ORR are present. Maybe authors should discuss further these data
Furthermore, is reported that 5-agents trials were more in the past decades than in recent years, while the number of 6-agents trial is the same. This could be an interesting aspect to discuss.
- line 204: this is not a paragraph. There is only the explanation of what ORR and synergism are, which is yet provided in the Materials and Methods section. Figure 5 and Table 1 are only cited but not introduced or analyzed/commented and this is not acceptable. Comment on Table 1 is present in the discussion section (lines 299 – 319) and should be moved here, but it won’t be sufficient for the reader to understand data in Table 1
- lines 234: “This indicates that the targeted agents…”, this statement cannot be inferred from the previous sentence, this sounds like an overstatement. Please provide a more clear and sustained reasoning
- lines 289 – 293: the first statement it’s unclear, probably not finished. All this part is confusing
- lines 318 – 319: authors claim that synergies need to be evaluated in vitro, but a large part of their results is based on the Kang et al. method which is used to infer synergistic or antagonistic combination of drugs used in clinical trial. So, authors analyze data with a method then, by the end of the discussion, say that a completely different approach (in vitro) is needed to understand agent combination effects. This is quite confusing. Pitfalls of the used methods need to be addressed differently
- line 326: “…the ORR of targeted agents is not a reliable surrogate endpoint for OS.” This is another pitfall of the presented study. For the previous, there is virtually no solution since authors are presenting a review and not a cellular biology study, but this problem could be resolved by using OS instead of ORR. Why do authors analyze clinical trials considering (and sometimes calculating, as stated in line 121) ORR instead of the more reliable OS? This is unclear, confounding and damages the reliability of the whole work
Minor issues:
- line 118: “…data were extracted from the most recent Phase II…”, please provide a date, “most recent” is not precise enough
- line 151: title of this paragraph is not clear
- Figure 4b: layout of this graph is different from the one of similar graphs shown in Figure 2b-c, please maintain consistency. More important, points referring to 1-agent and 4-agents trials miss the numbers of studies considered, which are present in other points
- Table 1: please insert a column containing the references of the works used to obtain the data for the calculation of the ORRo and ORRe
Author Response
Major issues:
- lines 123-124: authors declare to have used/calculated ORR based on RECIST which is a method (as declared in these lines by authors) relative to solid tumors. This implies that haematological cancers are not taken into consideration in this article. Maybe, this point needs to be clearly stated. Moreover, in Table 1, are cited Chronic Lymphocytic Leukemia and Non-Hodgkin Lymphoma. In these cases, probably RECIST criteria are referred to lymph nodes, but a clarification could be useful
Response: A distinction in the methods assessing response rate between solid and haematological cancer has been made to clarify ambiguity. The following sentence has been added:
“For haematological malignancies, Lugano response criteria41 and International Working Group’s revised response criteria42,43 were used to assess the ORR for lymphomas and leukaemia.” (line 166).
- line 190: authors constantly refer to a previous study (in this line and followings) but never provide a reference. It is mandatory to provide it
Response: The reference has been added.
- line 195: “… due to the lack of clinical data...”, actually the number of the reported studies with 5 agents are not so limited but the real problem could be the large difference between the old study and the modern data since they are, respectively 79 and 47. Likewise, trials with 6 agents are 81 in both studies so they are not limited nor different, nevertheless, no differences in ORR are present. Maybe authors should discuss further these data
Furthermore, is reported that 5-agents trials were more in the past decades than in recent years, while the number of 6-agents trial is the same. This could be an interesting aspect to discuss.
Response: Focus has been made on the 5-agent trials’ lower ORR replacing the insufficient clinical data argument (line 253). Also, this discrepancy was further explained in detail in the 8th paragraph of discussion.
Regarding the numbers of 5 and 6-agent trials in the modern dataset, the following sentence has been added in the discussion:
“However, the numbers of 5-agent and 6-agent trials are doubled and more than tripled, respectively, in our dataset (modern) compared to the older dataset (previous).” (line 470)
Also, the higher ORR of 6-agent combinations was explained in the same paragraph of discussion (line 477).
- line 204: this is not a paragraph. There is only the explanation of what ORR and synergism are, which is yet provided in the Materials and Methods section. Figure 5 and Table 1 are only cited but not introduced or analyzed/commented and this is not acceptable. Comment on Table 1 is present in the discussion section (lines 299 – 319) and should be moved here, but it won’t be sufficient for the reader to understand data in Table 1
Response: New sentences have been added to explain Figure 5. Also, separate paragraphs for Table 1 have been added which provide comments on the synergy and antagonistic list. Please note that the former Table 1 has been divided into two tables to identify synergy (Table 2) and antagonistic (Table 3) lists separately.
- lines 234: “This indicates that the targeted agents…”, this statement cannot be inferred from the previous sentence, this sounds like an overstatement. Please provide a more clear and sustained reasoning
Response: The sentence has been modified to make it sound clear and reasonable (line 304).
- lines 289 – 293: the first statement it’s unclear, probably not finished. All this part is confusing
Response: The statement has been rephrased to make it more readable and clearer (line 332).
- lines 318 – 319: authors claim that synergies need to be evaluated in vitro, but a large part of their results is based on the Kang et al. method which is used to infer synergistic or antagonistic combination of drugs used in clinical trial. So, authors analyze data with a method then, by the end of the discussion, say that a completely different approach (in vitro) is needed to understand agent combination effects. This is quite confusing. Pitfalls of the used methods need to be addressed differently
Response: In vitro work is required to understand mechanism of synergy/antagonism, precisely, which agents in the combination conferring synergy/antagonism to what extent. Therefore, Chou-Talalay’s Combination Index (CI) can quantify degree of conferred synergy/antagonism of each agent in the combination. Nonetheless, “mechanism and degree of synergy/antagonism” have been added in the sentence to remove the ambiguity.
- line 326: “…the ORR of targeted agents is not a reliable surrogate endpoint for OS.” This is another pitfall of the presented study. For the previous, there is virtually no solution since authors are presenting a review and not a cellular biology study, but this problem could be resolved by using OS instead of ORR. Why do authors analyze clinical trials considering (and sometimes calculating, as stated in line 121) ORR instead of the more reliable OS? This is unclear, confounding and damages the reliability of the whole work
Response: We did not analyse OS (overall survival) data because there are considerably a smaller number of Phase III clinical trials with OS data as opposed to abundant ORR and PFS data in Phase II trials. Moreover, OS data require longer time, approximately from a few years to 10 years to cover follow up treated patients to get OS data. Therefore, OS data would not reflect recent trends of the trials. On the flip side, ORR data is quick and even faster than PFS, and ideal for analysing a large number of Phase II studies.
Minor issues:
- line 118: “…data were extracted from the most recent Phase II…”, please provide a date, “most recent” is not precise enough
Response: A more specific timeline of the collected Phase II trials has been added (line 159).
- line 151: title of this paragraph is not clear
Response: Subtitle has been changed to “Impact of combination size”
- Figure 4b: layout of this graph is different from the one of similar graphs shown in Figure 2b-c, please maintain consistency. More important, points referring to 1-agent and 4-agents trials miss the numbers of studies considered, which are present in other points
Response: The mean ORR values for 1-agent and 4-agent have been added in the Figure 4b. However, maintaining consistency, for example, the Y-axis values of Figure 4b with the previous similar figures would make it vertically longer and the ORR trend line would be smaller. This would lessen the focus from the trend line. Therefore, we prefer the current layout of the Figure 4b.
- Table 1: please insert a column containing the references of the works used to obtain the data for the calculation of the ORRo and ORRe
Response: Two new columns have been added in Table 2 and Table 3 to refer the null model.
Reviewer 2 Report
The authors describe the analyzes of the synergistic and antagonistic combination of targeted and non-targeted cancer agents based on over 2k Phase trails. The researchers also studied how the trend of cancer treatment has changed over time.
Combination cancer therapies aim to improve the probability and magnitude of therapeutic responses and reduce the likelihood of acquired resistance in an individual patient. However, drugs are tested in clinical trials on genetically diverse patient populations.
The paper is interesting and contains valuable findings in regards to the combinatory therapies in cancer treatment.
The following points require further attention?
- Conventional chemotherapeutic and cytotoxic drugs were classified as non-targeted agents. In turn, synthetic hormonal therapies targeting specific receptor or receptors, monoclonal antibodies, targeted cancer drugs and immunotherapies including checkpoint blockers and CAR-T cell were classified as targeted agents. What about e.g. oncolytic viruses, did authors include trials studying their anticancer efficacy? A Table listing and describing representative of targeted and non-targeted agents is recommended.
- Did authors study the synergy between immunological agents like immune activators (oncolytic vectors, T cell therapy) and chemotherapeutics or the combinatory therapies between CPIs and immune activators?
- Also from the scientific point of view would be interested to know the interactions between various CPIs (synergy or antagonist). Did authors have a chance to investigate this combination?
- Are the Figures 2A and 2B presenting targeted or non-targeted therapies or their combinations?
- What is the difference between the modern and previous data set?
- Table 1 is showing that mainly chemotherapeutics or the combination of chemo plus enzymes/inhibitors can result in anti-cancer effect? What about the synergy between more than 2 agents?
- Please provide limitations of the study and made analyses.
- What could be the potential mechanism underlying observed synergy or antagonism?
Author Response
The following points require further attention?
- Conventional chemotherapeutic and cytotoxic drugs were classified as non-targeted agents. In turn, synthetic hormonal therapies targeting specific receptor or receptors, monoclonal antibodies, targeted cancer drugs and immunotherapies including checkpoint blockers and CAR-T cell were classified as targeted agents. What about e.g. oncolytic viruses, did authors include trials studying their anticancer efficacy? A Table listing and describing representative of targeted and non-targeted agents is recommended.
Response: Oncolytic viruses are a good point to consider. However, this type of agent did not appear in our unbiased selection of most recent 1,000 Phase II trials. Otherwise, we would have included oncolytic viruses in our study as a targeted agent. A new table (Table 1) simplifying targeted and non-targeted agents has been added in the methodology as recommended.
- Did authors study the synergy between immunological agents like immune activators (oncolytic vectors, T cell therapy) and chemotherapeutics or the combinatory therapies between CPIs and immune activators?
Response: In our synergy/antagonism study, we did not approach specific agent types, rather we analysed all agents in trials that were sufficiently available in our dataset as a single agent and their respective combinations in order to assess by the null model. Accordingly, yes, we found synergy/antagonism between immune activators such as CPIs with TKIs and chemotherapies. For example, the durvalumab (CPI) and ibrutinib (TKI) combination in non-Hodgkin lymphoma was reported as antagonistic in Table 3.
The other unreported combinations containing immunological agents including one CPI we found that are not statistically significant are as follows:
- Nivolumab + azacitidine,
- Vemurafenib + interleukin-2
- Also from the scientific point of view would be interested to know the interactions between various CPIs (synergy or antagonist). Did authors have a chance to investigate this combination?
Response: As explained above, identification of a particular synergistic/antagonistic interactions was entirely restricted by the availability of the trials testing that combination and their respective single-agent trials in our collected dataset. Although, two CPIs durvalumab and nivolumab with a TKI and chemotherapy, respectively, were evaluable by the null model and listed in the above point.
- Are the Figures 2A and 2B presenting targeted or non-targeted therapies or their combinations?
Response: Figure 2A and 2B comprises of all the agents in our collected 1,002 trials including targeted and non-targeted agents and their combinations.
- What is the difference between the modern and previous data set?
Response: The main difference between the two datasets is the time over which the trials were collected. The modern dataset (1,002 trials) contains most recent trials starting from the year 2020, whereas the previous dataset (1,163 trials) contains trials between 1990-2011.
As depicted in the Figure 4A, the impact of time has reflected in the proportion of the agent types (targeted vs non-targeted) in the modern and previous dataset.
- Table 1 is showing that mainly chemotherapeutics or the combination of chemo plus enzymes/inhibitors can result in anti-cancer effect? What about the synergy between more than 2 agents?
Response: We do not have enough data to analyse the interactions between more than 2 agents as it requires sufficient single agent trials of each individual drug in a combination. However, we were able to find out one combination containing more than 2-agent (FOLFOXIRI and Cetuximab). The combination of FOLFOXIRI (Leucovorin+Fluorouracil+Oxaliplatin+Irinotecan) and Cetuximab is listed in Table 3.
- Please provide limitations of the study and made analyses.
Response: The 10th paragraph of the discussion section addresses the limitations relating to the inferred synergy/antagonism and null model (line 492).
We recognise that addressing the limitations relating to synergy/antagonism in a separate paragraph is better comprehensible than combining them with the overall limitations of the study.
- What could be the potential mechanism underlying observed synergy or antagonism?
Response: Potential mechanisms for the synergistic combinations are as follows:
For the three chemotherapy combinations, destruction of the resistant tumour cells by a different killing pathway which could not be targeted by single drug alone. For rituximab and ibrutinib combination in CLL, ibrutinib impairs B cell development and maturation intracellularly, while rituximab binds to B cell’s surface and induces subsequent immune responses. These two distinct actions result in a robust anticancer response.
Potential mechanisms for the antagonistic interactions between a monoclonal antibody and a TKI is possibly due to failure of the antibody to induce an immune response and resistance of the TKI targeting specific intracellular signalling pathway. Antagonism between the chemotherapies could be due to targeting the same anticancer pathway or decreased chemotherapy sensitivity due to the recruited TAMs.
Round 2
Reviewer 1 Report
Changes are not well tracked, finding them in the text is difficult.
Major issues were poorly addressed like they were just minor suggestions.
Reviewer 2 Report
The authors provided satisfactory replies.